# 3DAxisPrompt: Promoting the 3D Grounding and Reasoning in GPT-4o

## Abstract

Multimodal Large Language Models (MLLMs) exhibit impressive capabilities across a variety of tasks, especially when equipped with carefully designed visual prompts. However, existing studies primarily focus on logical reasoning and visual understanding, while the capability of MLLMs to operate effectively in 3D vision remains an ongoing area of exploration. In this paper, we introduce a novel visual prompting method, called 3DAxisPrompt, to elicit 3D understanding capabilities of MLLMs in real-world scenes. More specifically, our method leverages the 3D coordinate axis and masks generated from the Segment Anything Model (SAM) to provide explicit geometric priors to MLLMs and then extend their impressive 2D grounding/reasoning ability to real-world 3D scenarios. Besides, we also provide a thorough investigation of the potential visual prompting formats and conclude our findings to reveal the potential and limits of 3D understanding capabilities in GPT-4o. Finally, we build evaluation environments with four datasets, *i.e.* ShapeNet, ScanNet, FMB, and nuScene datasets, covering various 3D tasks. Based on this, we conduct extensive quantitative and qualitative experiments, which demonstrate the effectiveness of the proposed method. Overall, our study reveals that GPT-4o, with the help of 3DAxisPrompt, can effectively perceive an object's 3D position in real-world scenarios. Nevertheless, a single prompt engineering approach does not consistently achieve the best outcomes for all 3D tasks. This study highlights the feasibility of leveraging MLLMs for 3D vision grounding/reasoning with prompt engineering techniques.

## 1 Introduction

In recent years, significant advancements and breakthroughs have been made in large language models (LLMs) (Brown et al., 2020; Chowdhery et al., 2022; Touvron et al., 2023; OpenAI et al., 2024). By aligning the representations with visual (and other) encoders, LLMs have been extended to multimodal large language models (MLLMs[1]) (GeminiTeam, 2024; OpenAI, 2024), which are capable of handling richer visual modalities. These studies have attracted significant interest from researchers, with numerous works continuously being proposed to enhance the reasoning capabilities of MLLMs in various aspects. For example, Yang et al. (2023a) leverage the SoM prompting to enable the visual grounding of GPT-4v, and Wu et al. (2024b) achieve accurate object detection with MLLMs in a Chain-of-Thought (Wei et al., 2023) manner. By leveraging the advanced reasoning capabilities of the language model component, MLLMs have been explored for perception and interaction with a variety of applications.

However, existing MLLMs are mainly pretrained with 1D data (*e.g.* texts) and 2D data (*e.g.* images), while the real-world challenges are inherently spatial and require spatial grounding in the context of 3D scenes. In this context, a critical question emerges:

*Do vision-language-based MLLMs possess the capability for 3D grounding and reasoning?*

Although lots of studies have explored the application of MLLMs in 3D scenarios, these works have not directly leveraged the 3D grounding and reasoning capabilities of MLLMs. For example, some work (Wen et al., 2024; Cui et al., 2024) apply MLLMs in the field of autonomous driving, however, they primarily leverage MLLMs for decision-making rather than 3D scene understanding.

---

[1]also known as large multimodal models (LMMs).

Besides, PointLLM (Xu et al., 2023) empowers MLLMs to understand 3D point clouds with additional point-text instruction training. Although this solution aligns high-level representations of points and texts, it only supports specific comprehension tasks (*e.g.* classification and captioning for single 3D objects) and does not truly activate the fine-grained 3D perception capabilities of MLLMs. Overall, existing studies do not fully answer the question we raised, and further in-depth exploration is required.

In this paper, we aim to investigate how to extend the exceptional 1D/2D grounding and reasoning capabilities of MLLMs into the 3D world space without further fine-tuning. Based on this requirement and the inspiration of visual prompts, *e.g.* (Yang et al., 2023a; Wu et al., 2024b), we propose a new prompting mechanism, called 3DAxisPrompt. Specifically, given the point cloud of a real scene, we first embed the 3D coordinate axis and meshes in this scene in an automatic manner to provide the 3D geometric priors. Then calibrated scene will be rendered into observation images from different angles. Furthermore, to introduce object-level semantic cues, we overlay the masks generated by the Segment Anything Model (SAM) (Kirillov et al., 2023) with numerical or alphabetic marks, similar to SoM (Yang et al., 2023a). In this way, we can extend MLLMs impressive 2D grounding/reasoning capabilities to real-world 3D scenarios.

Besides, for the first time, we present a comprehensive exploration of potential visual prompt formats, such as coordinate axis, masks, bounding boxes, marks, color highlights, *etc.*, in MLLMs for 3D understanding. Based on our investigation, we also conclude some findings to reveal the potential and limits of 3D understanding capabilities in MLLMs. For example, multi-view visual prompting cannot directly activate the 3D reasoning capabilities of MLLMs, but tri-view prompting can. Finally, we construct evaluation environments using four datasets—ShapeNet, ScanNet, FMB, and nuScene—covering a range of 3D tasks. We then conduct extensive quantitative and qualitative experiments, demonstrating the effectiveness of the proposed approach.

Overall, our objective is not to achieve perfect zero-shot performance with GPT-4o, but to explore its limitations and potential in zero-shot inference for 3D grounding/reasoning. We expect that future improvements to the MLLMs will lead to further quantitative gains on the actual tasks. To summarize, our main contributions are:

- We propose a visual prompt scheme called 3DAxisPrompt. By inserting the 3D coordinate axis in a real scene, the proposed 3DAxisPrompt can elicit the 3D grounding and reasoning capabilities in GPT-4o, such as 3D localization and planning.

- We provide the first comprehensive investigation of the potential visual prompt formats of MLLMs for 3D understanding. Besides, we conclude our findings to reveal the potential and limits of 3D understanding capabilities in GPT-4o.

- We conduct extensive experiments on a wide range of tasks, including indoor and outdoor 3D localization, route planning, and robot action prediction. These results demonstrate the proposed 3DAxisPrompt can effectively enhance 3D understanding capabilities in GPT-4o.

## 2 RELATED WORK

**LLMs and MLLMs.** Significant progress has been witnessed in LLMs (Chowdhery et al., 2022; Touvron et al., 2023; Zhang et al., 2022; OpenAI et al., 2024). Trained on internet-scale data, LLMs are effective commonsense reasoners (Zhao et al., 2023). MLLMs (Liu et al., 2023a; Lu et al., 2024; Bai et al., 2023) integrate vision encoders (Radford et al., 2021) into LLMs, allowing them to reason over visual input directly. State-of-the-art MLMMs like GPT-4V, Gemini (GeminiTeam, 2024), Claude (The), and GPT-4o (OpenAI, 2024) have excelled in general vision-language tasks (Wu et al., 2023; Yang et al., 2023c; Fu et al., 2023). Leveraging the advanced vision-language reasoning ability, the exploration has been made of MLMMs in perception and interacting with the physical world (Lu et al., 2024), including autonomous driving (Wen et al., 2024; Cui et al., 2024), anomaly detection (Cao et al., 2023), robotic control and learning (Collaboration et al., 2024; Brohan et al., 2023), which requires fine-grained 3D spatial grounding that remains to be explored (Chen et al., 2024). To promote the connection of MLMMs to the real physical world (Chen et al., 2024), we aim to find a strategic prompting method to elicit and promote the 3D grounding and reasoning in MLMMs regarding a real 3D world, such as to reason about the 3D location of an object.

**Visual prompting.** Prompt engineering has emerged as a promising approach to improve MLLMs across multiple domains, such as in-context learning (Brown et al., 2020; Dong et al., 2024), Chain-of-Thought and Tree-of-Thought (Wei et al., 2023; Yao et al., 2023). Consequently, numerous prompting methods have been developed to improve visual grounding in MLLMs. Colorful prompting tuning (CPT) (Yao et al., 2022) overlays color-based co-referential markers in both images and text and enables strong few-shot and even zero-shot visual grounding capabilities. RedCircle (Shtedritski et al., 2023) guides the vision model to an enclosed region by adding a red circle. Blur Reverse Mask (Yang et al., 2023b) blurs the area outside the target mask to leverage the precise mask annotations to reduce focus on weakly related regions while retaining spatial coherence. These two methods promote fine-grained visual grounding. Furthermore, Lei et al. (2024) enhances the vision-language coordination by SCAFFOLD prompting that scaffolds coordinate on images. Set-of-Mark (Yang et al., 2023a) add a set of visual marks on top of image regions. Both these two methods indicate the emergent 2D spatial grounding (Mitra et al., 2024; Islam et al., 2023) in MLLMs, including 2D position and relation inference. To enhance the 3D spatial grounding, Nasiriany et al. (2024) propose an iterative prompting method (PIVOT) to infer the robot action considering spatial relation. COARSE CORRESPONDENCES (Liu et al., 2024) prompts the MLLMs to elicit the 3D spacetime understanding. These two methods concentrate on 3D spatial relation instead of 3D spatial position, showing limited performance in instance-level tasks that demand precise 3D localization and recognition. Our study strives to extend the 2D spatial grounding (Lei et al., 2024; Yang et al., 2023a) to 3D grounding by formulating a visual prompting method, promoting spatial position inference in MLLMs.

**GPTs and grounding.** Generative Pretrained Transformers (GPTs) (Brown et al., 2020; OpenAI et al., 2024) have led to a breakthrough in the realm of natural language processing. As a leading LMM, GPT-4V has significantly expanded the boundaries of MLLMs capabilities and shown abilities to understand visual annotations (Yang et al., 2023c) and solve visual reasoning tasks, such as web navigation (Yan et al., 2023a; Zheng et al., 2024), autonomous driving (Wen et al., 2024; Cui et al., 2024), and medicine diagnostics (Yan et al., 2023b; Liu et al., 2023b). Furthermore, GPT-4o (OpenAI, 2024) is the latest development in a string of innovations to MLLMs, which has shown significant performance in multiple tasks (Joe et al., 2024; Wu et al., 2024a; Shahriar et al., 2024; Hu et al., 2024). Our study is to explore a prompting method to promote the 3D spatial grounding in MLLMs. Since GPT-4V is proven to outperform the other models in visual grounding when equipped with visual prompts (Yang et al., 2023a) and GPT-4o shows significant improvement in 3D spacetime understanding (Liu et al., 2024), we believe the GPT-4o can present representative 3D spatial grounding abilities in MLLMs and conduct our experiments and analysis using the GPT-4o.

# 3 3DAXISPROMPT

Unlike previous 2D visual prompts that primarily focus on planar object relationships, we aim to introduce a 3D prompts method to enable effective 3D spatial reasoning and grounding in GPT-4o for real 3D environments. In this section, We revisit approaches for incorporating 3D information into visual prompts and propose an effective method 3DAxisPrompt, to enhance 3D spatial information through visual prompts.

## 3.1 PROBLEM FORMULATION

The goal of 2D visual prompts is to enhance the MLLMs's understanding of visual information by adding auxiliary information to the original images. This can be expressed by the following equation:

$$T^o = \mathcal{F}(T^i, VP(I)), \tag{1}$$

where $T^o = \left[t_1^o, \ldots, t_{l_o}^o\right]$ represents textual output with a length of $l_o$ from a foundation multimodal language model $\mathcal{F}$. This output is generated given a task textual description $T^i$ and a visual prompt $VP(I)$ derived from an observation image $I$.

However, directly annotating and representing real 3D scenes is a more demanding task compared to 2D prompts, as it requires consideration of spatial depth, occlusions, and intricate object relationships (Liu et al., 2024). A common approach is to utilize multiview images instead of original 3D representations while adding corresponding annotations to the 2D images. Since GPT-4V has

been shown to significantly outperform other MLLMs in grounding ability when visual prompts are added (Yang et al., 2023a), we employ GPT-4o as $\mathcal{F}$ in this work.

Meanwhile, unlike previous visual prompt approaches that solely add 2D spatial information, we discovered that when GPT-4o is challenged with both the point cloud $p^i$ provided in text format and visual prompts, it can recognize the text file as the point cloud $p^i$ and reason about spatial positions based on the input sequence $T^i$, the observation image $I$, and the point cloud $p^i$.

We experimented with various visual prompt formats to determine the optimal way to transform an input image $I$ into a marked image $I^m$ with 3D cues. After evaluating different 3D cue representations through spatial reasoning tasks, we propose the 3DAxisPrompt framework, as illustrated in Figure 1.

Given a point cloud as input, 3DAxisPrompt adds the 3D axis to the point cloud and renders observation images from multiple views of the point data. For each view, SAM (Kirillov et al., 2023) is used to highlight the boundary of the region of interest and overlay the mark. Consequently, the observation $I$ becomes an image sequence $I_j^m = [I_1^m, \ldots, I_j^m]$. Formally, Equation 3.1 becomes:

$$T^o = \mathcal{F}(T^i, p^i, \underbrace{3DAxis(I)}_{I_j^m}). \tag{2}$$

By incorporating the 3D axis and overlaying marks and contours onto the rendered observation image of a point cloud, the 3DAxisPrompt enables GPT-4o to perform 3D spatial grounding tasks such as localization, route planning, and robot action prediction.

In the following sections, we will delve into exploring the impact of adding various 3D visual cues on GPT-4o's spatial grounding and reasoning capabilities. Subsequently, we will conduct quantitative experiments to assess the performance of the proposed 3DAxisPrompt framework.

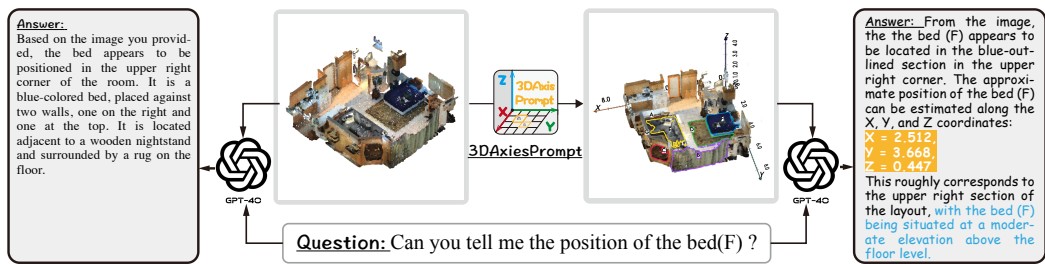

Figure 1: Comparing standard GPT-4o and its combination with 3DAxisPrompt. It shows that the proposed 3DAxiesPrompt helps GPT-4o to reason about the 3D spatial position. We highlight the differences between our method and the standard one.

## 3.2 INVESTIGATION ON ENCODING 3D CUES

Visual prompts, such as marks (Yang et al., 2023a; Liu et al., 2024), masks (Yang et al., 2023b), colors (Yao et al., 2022), scaffolding points (Lei et al., 2024), arrows (Nasiriany et al., 2024), and red circles (Shtedritski et al., 2023), have been shown to provoke 2D spatial grounding in GPT-4o. These visual prompts can be seen as integrating spatial information into images for grounding in image-text pairs (Brown et al., 2020; Li et al., 2022), leading to 2D spatial grounding. To extend 2D spatial grounding to 3D space, we propose encoding additional 3D cues into observation images to trigger 3D perception in GPT-4o. Based on this, we explore effective methods for representing these 3D cues.

**3D axis integration in scenes.** We found that adding a 3D axis to the point cloud of a 3D instance and rendering observation images with the $x$, $y$, and $z$ axes as visual prompts enables GPT-4o to reason about 3D positions, as shown in Figure 2. This approach allows GPT-4o to associate semantics with spatial locations defined by the 3D axis, thereby facilitating 3D spatial grounding.

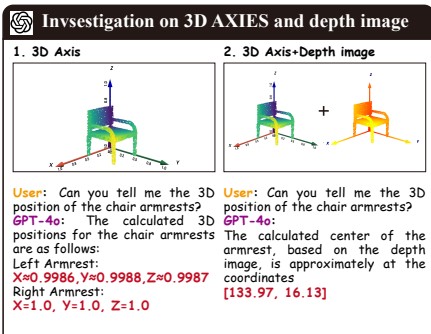 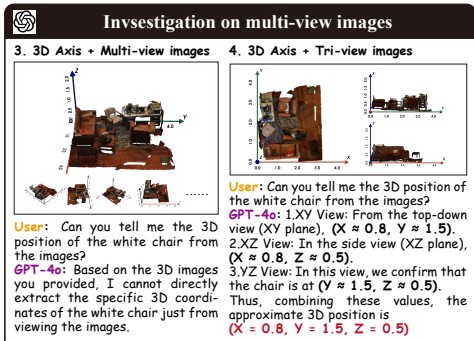

Figure 2: Investigation on encoding 3D cues in visual prompts. We present some examples of the investigations on the 3D Axis, depth image, multi-view images, and tri-view images. Depth image and multi-view images fail to provoke the 3D spatial position inference.

**Depth compensation.** Although 3D Axis prompts enable basic spatial grounding, the spatial positions inferred by GPT-4o lack accuracy, especially along the depth direction. We further explore potential solutions to compensate for the missing dimensions, including leveraging RGB-D images as visual input, as shown in Figure 2. More results are presented in Appendix A1. In conclusion, none of these depth compensation methods yielded satisfactory results. While GPT-4o can recognize depth images and surface color as depth or distance information, the depth and 2D positions are predicted separately, indicating a lack of interaction between them.

**3D coordinates information.** Based on our findings during depth compensation, we believe that encoding all 3D information solely within visual prompts is overly challenging (Liu et al., 2024). Additional 3D cues are necessary beyond just visual prompts. Furthermore, we discovered that GPT-4o can recognize point clouds formatted as coordinates in the input text, as demonstrated in Appendix A. However, when these points are combined with a 3D Axis visual prompt, GPT-4o effectively incorporates them for reasoning about 3D spatial positions. Consequently, we consider the point cloud in text format to be an essential input for the model.

**Multiview and tri-view images.** Inspired by Structure from Motion (SFM) (Schönberger & Frahm, 2016), which can reconstruct 3D structures from a series of 2D images, and tri-plane methods (Shue et al., 2023), which decompose a 3D scene into three distinct 2D projections, we further investigate the multiview and tri-view images of an actual scene. As shown in Figure 2, we render the images with the 3D axis of the actual scene from different angles. Additional results are provided in Appendix A1. Our findings indicate that the multi-view image sequence can only trigger 3D spatial grounding in GPT-4o when combined with text-formatted point clouds. In contrast, the tri-view images successfully provoke 3D spatial grounding in GPT-4o even without the text-formatted point cloud input. However, when reasoning about complex scenes, tri-view encounters significant occlusion issues, leading to considerable inaccuracies.

Based on these aforementioned findings, we incorporate the 3D Axis into the 3D scene and render observation images from various angles as the visual prompts.

### 3.3 INVESTIGATION ON MARK FORMATS

We explored two methods for overlaying marks on visual prompts. The first method involves adding 2D marks directly onto the observation image, while the second method inserts 3D marks into the 3D space and then renders the observation image with these marks.

**2D marks.** The 2D marks are obtained using SAM to segment the objects of interest in the observation image. We consider two types of 2D marks: those on top-view images and those on perspective images, as illustrated in Figure 3. We also evaluate four main 2D mark formats—point, polygon, mask, and bounding box (see Appendix A2). Our empirical study indicates that all mark formats, when combined with the 3D Axis, successfully elicit 2D spatial grounding in GPT-4o.

**3D marks.** For 3D marks, we investigate the use of 3D bounding boxes and 3D edge points, as shown in Figure 3. We evaluate four types of 3D markers: marks, Axis-Aligned Bounding Boxes

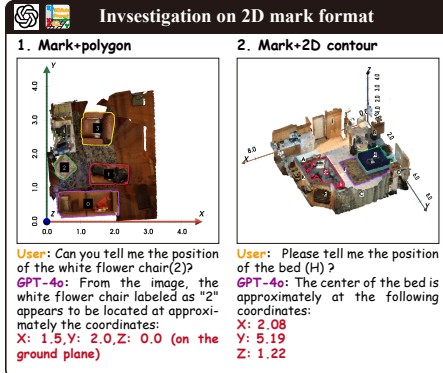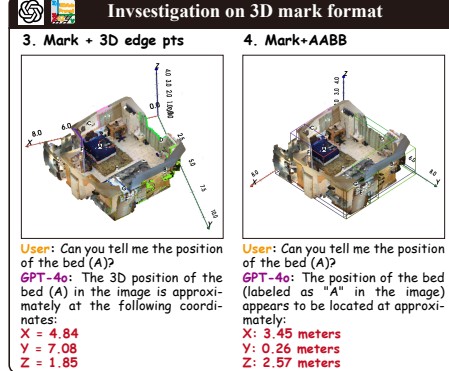

Figure 3: Some examples of the investigation on 2D and 3D mark formats. All the mark formats successfully provoke the 3D spatial position reasoning.

(AABB), Oriented Bounding Boxes (OBB), and 3D edge points. The 3D edge points are filtered from the input point cloud based on their normals. The visual results demonstrate that all the 3D marks successfully elicit 3D spatial grounding in GPT-4o. Additional results are provided in Appendix A3.

In conclusion, using both 2D and 3D marks in visual prompts can effectively elicit 3D spatial position reasoning in GPT-4o. To determine the optimal mark format, we evaluate all the mark formats in the following section. The quantitative results indicate that both the combination of (mark + 3D edge points) and (mark + 2D contour) perform better than the others, with the 2D contour outperforming the 3D edge points. This underscores the importance of object contours in visual prompts for 3D spatial position reasoning. Additionally, we employ multiview images instead of tri-views to mitigate the occlusion problem.

## 4 EXPERIMENTS

### 4.1 EXPERIMENTAL SETUP

**Implementation.** Our method does not require model training. However, due to the limited and costly GPT-4o API quota, we must exhaustively send 3DAxisPrompt-augmented images to the Chat-GPT interface. To efficiently manage experiments and evaluations, we employ a divide-and-conquer strategy, opening a new chat window for each scene to prevent context leakage. All reported results are obtained in a zero-shot manner.

**Benchmarks.** Given the limited GPT-4o quota, we could not fully evaluate the validation set for each task. Instead, we randomly selected 20 scenes from each test dataset as validation data. We aimed to cover as many diverse scenes as possible across all datasets to preserve their original diversity. For each instance, we applied the 3DAxisPrompt to the observation images of the point cloud using our custom toolbox.

### 4.2 QUANTITATIVE RESULTS

**Indoor localization.** On the indoor localization task (shown in Figure 1), we evaluate the localization errors of the 3DAxisPrompt on the subset of the Scannet (Dai et al., 2017) to fully analyze mark formats shown in Figure 3. Also, we integrate the Chain-of-Thought (CoT) (Mitra et al., 2024) with the proposed 3DAxiesPrompt and provide the additional coordinate of a nearby object to append *let's think step by step*. No previous work has presented localization errors related to 3D spatial grounding. We use the Normalized Root Mean Squared Errors (NRMSE) to quantify the spatial localization errors, as defined in Equation 4.2:

$$\text{NRMSE} = (\sum_{j=1}^{N} \frac{\sum_{i=1}^{n_j} \mathcal{D}(\hat{x}_i, x_i)}{n_j \cdot max(x_i)})/N \tag{3}$$

where $\hat{x}_i$ is the predicted position of the object $i$ in scene $j$ while $x_i$ is the ground-truth position. $n_j$ is the total number of the objects in scene $j$, and $N$ is the total number of scenes selected for evaluation. $\mathcal{D}$ is the function to measure the distance between the predicted position $\hat{x}_i$ and the ground-truth position $x_i$. Two types of distance measurement function $\mathcal{D}$ are selected, including the distance to the object center (To center) and the distance to the bounding box (AABB) (To bbx). We use the Euclidean distance to measure the to-center distance. As for the to-bbx distance, we calculate the minimum distance from the predicted position to the AABB of the object.

Table 1: Main quantitative results of indoor localization on ScanNet dataset.

| Mark Type | Prompt Elements | ScanNet | |
| --- | --- | --- | --- |
| | | To center | To bbx |
| 3D Mark | Mark | 0.333 | 0.216 |
| | Mark+OBB | 0.350 | 0.231 |
| | Mark+AABB (red) | 0.376 | 0.219 |
| | Mark+AABB (colors) | 0.311 | 0.207 |
| | Mark+3D edge points | 0.305 | 0.205 |
| 2D Mark | 2D contour (colors) | 0.320 | 0.175 |
| | **Mark+2D contour (colors)** | **0.271** | **0.138** |
| | **Mark+2D contour (colors) + CoT** | **0.219** | **0.115** |

We present the quantitative results of the indoor localization in Table 1. It can be seen that besides the CoT, the combination of the mark and 2D contour achieves the best performance with a 7% decline in to-bbx distance errors compared to the Mark visual prompts. When combined with the CoT, the 3DAxiesPrompt achieves a 19% improvement on to-center distance. In the 3D mark, the (mark + 3D edge points) outperforms the others, and the performance of the OBB, AABB, 3D edge points, and 2D contour gradually improves. The bounding box, 3D edge point, and contour are the same in some ways because they all intend to depict the boundary of each instance region in a scene. This rend shows the importance of highlighting the instance boundary in visual prompts. Also, compared to a single color (red), highlighting each object boundary using different colors sees a 7% decline in to-center distance errors.

Table 2: Quantitative results of route planning, outdoor localization, and robot action prediction.

| Task | Specification | ScanNet | nuScenes | | FMB |
| --- | --- | --- | --- | --- | --- |
| | | Success rate | To center | To bbx | Success rate |
| Route Planning | From door to chair | 80% | n/a | n/a | n/a |
| | From door to bed | 100% | n/a | n/a | n/a |
| | From door to desk | 70% | n/a | n/a | n/a |
| | From couch to bed | 90% | n/a | n/a | n/a |
| | From door to chair | 60% | n/a | n/a | n/a |
| | Average | 79% | n/a | n/a | n/a |
| Outdoor Localization | Vehicle | n/a | 0.306 | 0.165 | n/a |
| | Vegetation | n/a | 0.283 | 0.143 | n/a |
| Robot Action Prediction | Grasp | n/a | n/a | n/a | 72.5% |
| | Release | n/a | n/a | n/a | 62.5% |

**Route planning.** We evaluate the indoor route planning abilities on the subset of Scannet (Dai et al., 2017), as shown in Figure 4. Following PIVOT (Nasiriany et al., 2024), we evaluate the performance via whether the navigation successfully reaches the destination. We select some common tasks that happen frequently in real life.

The quantitative results of the route planning task are demonstrated in Table 2. The 3DAxisPrompt achieves an average success rate of 79%, proving that the indoor route planning ability in GPT-4o is equipped with 3DAxisPrompt. However, when encountered with objects densely located together, such as chairs, the 3DAxisPrompt is more likely to fail (70% in from door to desk when needing to detour past many chairs).

**Outdoor localization.** We evaluate the outdoor localization of the 3DAxisPrompt on the subset of the nuScenes (Caesar et al., 2020) dataset. The point cloud is very sparse in nuScenes, so we choose the two types of obstacles frequently encountered in autonomous driving, including vehicle and vegetation, as shown in Figure 5. We use the same merits defined in Equation 4.2 to quantify the performance.

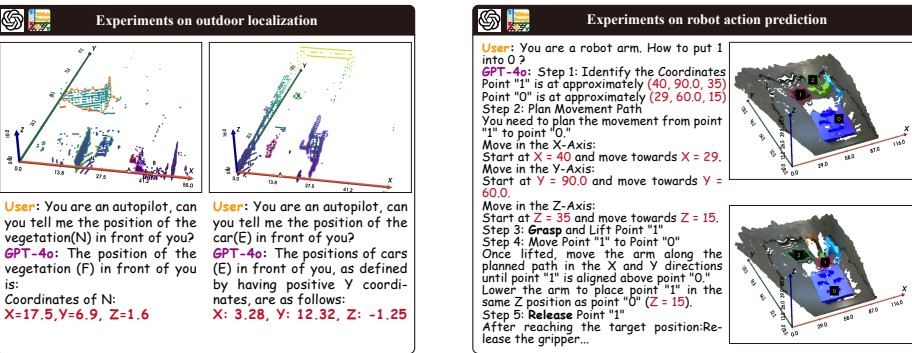

Figure 4: Experiments on route planning. It shows that our method 3DAxiesPrompt helps GPT-4o to plan the route based on spatial localization. We highlight the differences between our method and the standard one.

Figure 5: Some examples of the experiments on outdoor localization and robot action prediction.

The quantitative results of the outdoor localization are shown in Table 2. The localization performance of the vehicle is better than that of the vegetation.

**Robot action prediction.** In addition to localization and navigation tasks, we also examine the 3DAxisPrompt for robot action prediction on the subset of the robot control dataset FMB (Luo et al., 2024). There is no point data in the FMB, so we transform the RGBD images to point clouds according to the camera intrinsic as the evaluation data, as shown in Figure 5. The task is to predict the action to place the object onto the target destination, assuming the GPT-4o is a robot arm. Two types of actions are evaluated separately, namely grasp and release, because these two actions are the central part of the robot's grasping task. We evaluate the performance by determining whether the orders can complete the mission.

Table 2 presents the quantitative results. Equipped with the 3DAxisPrompt, GPT-4o can complete simple robot action prediction tasks.

**Coarse object generation.** We also evaluate the 3DAxiesPrompt for coarse object generation task on Shapenet (Chang et al., 2015) dataset, as shown in Figure 6. Some keypoints of an object are marked and predicted using the 3DAxiesPrompt. Then, a coarse object skeleton is constructed based on the answers.

## 4.3 ABLATION STUDY

We conduct an ablation study on elements that may affect the GPT-4o to 'read' the coordinates from the 3D Axis, including the number of muti-view images and the axis elements.

**The number of images.** We conduct the ablation study on the number of observation images through the indoor localization tasks on the subset of the Scannet (Dai et al., 2017) dataset. The experimental results are shown in the line graph of Figure 7. A trend can be noticed that by increasing the number of scene views, the localization errors gradually decrease. The eight observation images outperform the others and achieve a 41% improvement compared to a single image.

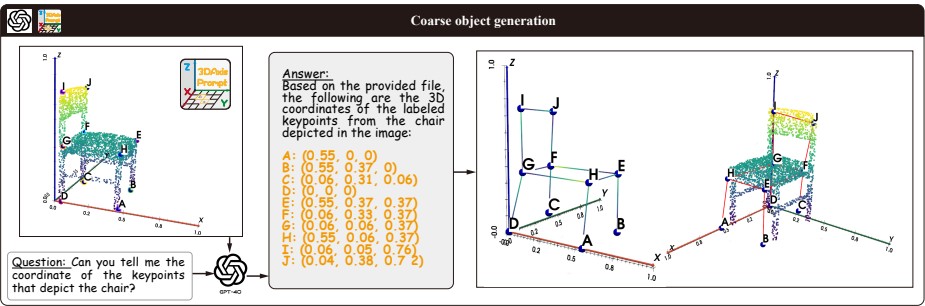

Figure 6: Coarse object generation on Shapenet dataset. It shows that based on our method, GPT-4o can reason about the keypoints that can represent the skeleton of an object.

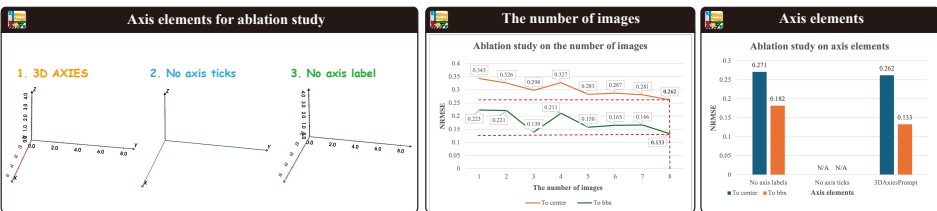

Figure 7: The axis elements considered for ablation study and the results of the number of images and the axis elements.

**Axis elements.** The elements of the axis, including the axis ticks and labels, are studied as shown in Figure 7. From the quantitative results shown in the histogram of Figure 7, we can see that the 3DAxisPrompt fails to provoke the spatial position reasoning without the axis ticks. Also, the axis label is essential, without which the errors of the to-bbx distance will increase by 37%.

## 5 DISCUSSION AND CONCLUSION

**Where dose the 3D spatial grounding comes from?** Our understanding is derived from experimental observations. We hypothesize that the 3D Axis offers essential scale information and spatial cues that serve as a foundation for localization. Interestingly, even without the 3DAxisPrompt, GPT-4o can make rough estimates of distances between objects when provided with an observation image of a real scene. However, by incorporating the 3D Axis, these estimates become more precise, as the axis ticks unify the units of measurement, allowing for a more accurate perception of distance. Additionally, the axis origin and direction act as reference points, supporting the localization process. In this way, the 3DAxisPrompt reinforces 3D spatial grounding by offering crucial 3D cues.

**The essential factors in 3DAxisPrompt.** The axis ticks and the highlighted contour of an object in the observation images are essential in 3DAxisPrompt. More specifically, the axis ticks provide an essential ruler to measure the world, while the contours marked in the observation images can significantly improve the localization performance. Also, we find that the localization performance can be further enhanced if given the precise coordinates of the objects (reference points) around the queried one. We think this is the same as human perception; the additional reference point makes the coordinate easier to read.

**Conclusion.** In this paper, we propose a visual prompt scheme called 3DAxisPrompt for MLLMs, particularly GPT-4o, aimed at enhancing 3D spatial grounding. By overlaying visible 3D axis, markers, and region edges on observation images from different angles, 3DAxisPrompt enables tasks like localization and spatial reasoning. Our study shows how various 3D visual prompts help GPT-4o interpret 3D space, with qualitative results indicating fine-grained perception and reasoning in real-world scenarios. We hope this work inspires future research on applying MLLMs to real-world interactions and advancing AI in everyday life.

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

## A APPENDIX

### A1 INVESTIGATION ON ENCODING 3D CUES

We present more investigation of encoding 3D cues in the visual prompts, as shown in Figure A1. It can be noted that GPT-4o can identify the distance information encoded on the object surface, but this extra information is not used to reason about spatial location. Also, the points.txt can be understood only if with the 3D Axis visual prompts. Both the RGB-D and depth information are not integrated into spatial reasoning.

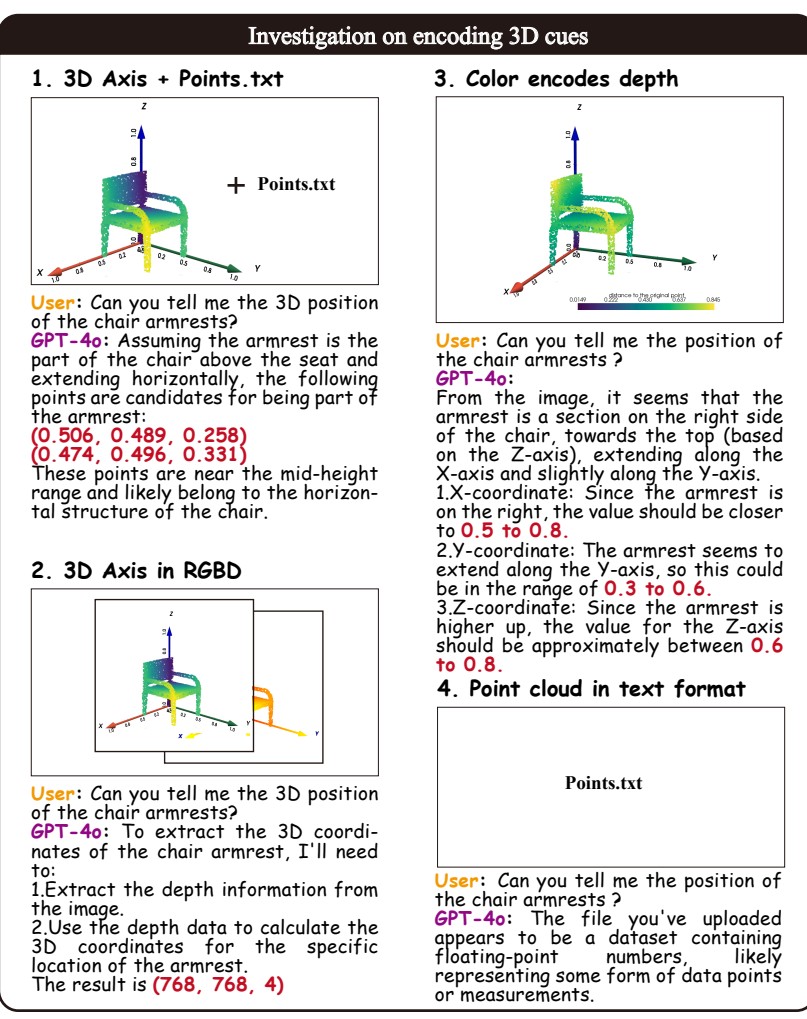

Figure A1: Investigation on encoding 3D clues in visual prompts.

We present more investigation of multi-view images, as shown in Figure A2. The multi-view images can only promote the sptial localization with the points.txt file. The tri-view images can promote spatial localization, but the objects are easy to block, as shown in Figure A2. Voxel can also represent a 3D scene, but it defects the spatial localization.

### A2 INVESTIGATION ON 2D MARK FORMAT

We present more investigation of 2D mark formats, as shown in Figure A3 and A4. All the 3D mark formats can promote the spatial localization.

## A3    INVESTIGATION ON 3D MARK FORMAT

We present more investigation of 3D mark formats, as shown in Figure A5. All the 3D mark formats can promote the 3D spatial localization.

## A4    LIMITATIONS

Even though the evaluation proves that the 3DAxisPrompt can promote the 3D spatial grounding in GPT-4o on some tasks, we have to admit that the performance is not perfect. When the objects are too small to be identified, or the boundaries are not clear enough, the performance will significantly drop. Moreover, we find that the GPT-4o still struggles to read the information encoded in the 3D Axis when the objects are far away from the 3D Axis.

## Investigation on multi-view images

### 1. 3D Axis+Tri-view+Points.txt

**User**: Can you tell me the 3D position of the white chair from the images?
**GPT-4o**: Final Estimated 3D Position of the White Chair:
**X ≈ 0.5 meters**
**Y ≈ 3 meters**
**Z ≈ 0.8 to 1 meter**

### 2. Voxel

**User**: Can you tell me the 3D position of the white chair from the images ?
**GPT-4o**: The image you uploaded appears to represent a 3D voxel map or a block-style rendering. You asked for the position of a white chair in the 3D space shown, but without additional details on how this 3D structure is defined or segmented, it's hard to extract precise coordinates directly from the image alone.

### 3. Voxel + multi-view images

**User**: Can you tell me the 3D position of the white chair from the images ?
**GPT-4o**: To determine the 3D position of the white chair from these images, I would need a method to compute it based on the image data, such as image processing techniques for identifying the chair in each image and using the position from different angles to triangulate the 3D position.

### 4. 3D Axis+Multi-view+Points.txt

**User**: Can you tell me the 3D position of the white chair from the images?
**GPT-4o**: Based on the data from the text file, one of the likely 3D positions for the white chair could be found from the following coordinates:
**X: 1.884, Y: 3.538, Z: 1.803**

### 5. Voxel+multi-view+points.txt

**User**: Can you tell me the 3D position of the white chair from the images ?
**GPT-4o**: In the 3D views, we can use the X, Y, and Z axes to describe the chair's position. It appears to be in the lower left region in most of the views. From the images:
In the top-down view (image 2), the white chair is roughly around the coordinates
**X: 0.5 - 1.0, Y: 2.0 - 2.5.The Z-axis height seems to be around 0.5 in several views (images 1 and 6).**

### 6. Voxel + Points.txt

**User**: Can you tell me the 3D position of the white chair from the images ?
**GPT-4o**: Based on the data in the file, the white chair seems to be positioned around the following 3D coordinates:
**x=0.471**
**y=0.491**
**z=0.311**

Figure A2: Investigation on multi-view images.

Figure A3: Investigation on 2D marker format (I).

Figure A4: Investigation on 2D marker format (II).

## Investigation on 3D mark

### 1. Mark

**User**: Can you tell me the position of the bed (A)?
**GPT-4o**: The 3D position of the bed (marked as A) is approximately located at coordinates
X = 0.0, Y = 2.3, Z = 1.7,
based on the data from the provided 3D scene file

### 2. Red AABB

**User**: Can you tell me the coordinate of the bed (E) ?
**GPT-4o**: The coordinates for the bed, labeled as "E" in the image, are approximately:
(3.08, 1.49, 0.76).
These coordinates are extracted from the dataset based on object "E". It seems to refer to a point near the middle of the bed, with the dimensions indicating a placement within the room.

### 3. Red OBB

### 4. Mark+OBB

**User**: Can you tell me the position of the bed (A)?
**GPT-4o**: The 3D position of the bed (A) in the scene is approximately located at coordinates
X = 3.7, Y = 6.67, and Z = 1.22

### 5. 3D Edge points

**User**: Can you tell me the position of the bed (H) ?
**GPT-4o**:
In the image, point H is visually located on the bed, in the upper right corner near the coordinate axes. By cross-referencing the scene data file, it seems the relevant line for point H could correspond to:
X = 5.96
Y = 2.35
Z = 0.17

**User**: Please tell me the position of H?
**GPT-4o**: The three-dimensional coordinates of object H are approximately:
x=5.299
y=0.634
z=1.448
These coordinates indicate the position of object H within the scene.

Figure A5: Investigation on 3D marks.

