# OpenReview forum: "3DAxisPrompt: Promoting the 3D Grounding and Reasoning in GPT-4o"
_ICLR.cc/2025/Conference — Submitted to ICLR 2025_

### Official Review · Reviewer_HR2p · 2024-10-27

**Soundness:** 3
**Presentation:** 4
**Contribution:** 3
**Rating:** 6
**Confidence:** 4

**Summary:**

This paper presents a novel approach to enhance the capabilities of Multimodal Large Language Models (MLLMs) in understanding and reasoning about 3D environments. The authors propose a new prompting mechanism, 3DAxisPrompt, which integrates 3D geometric priors by embedding 3D coordinate axes and meshes into the scene. This method aims to extend the existing 2D grounding and reasoning capabilities of MLLMs into the 3D space without requiring further fine-tuning.
The paper discusses various visual prompting techniques and their effectiveness in improving visual grounding, highlighting the limitations of current methods in fully activating the fine-grained 3D perception capabilities of MLLMs. The authors conduct comprehensive evaluations using different datasets to assess the performance of their proposed method, demonstrating its potential to enhance the understanding of spatial relationships in 3D contexts.

**Strengths:**

1. The proposed visual prompting method 3DAxisPrompt is novel and has been proven to enhance the 3D understanding capabilities of MLLMs. With the proposed technique, the MLLM can be directly adapted to localization, route planning, and action prediction without the need for additional training.
2. The authors conducted extensive quantitative and qualitative experiments across multiple datasets (ShapeNet, ScanNet, FMB, and nuScene), to show the effectiveness of their proposed method in various 3D tasks.
3. The methodology is well-defined and clearly stated, which allows for reproducibility and further investigation by peer researchers.

**Weaknesses:**

1. How are the 3D axes set? Does the origin always lie in the same corner of the scene?  What if the axes are not well aligned with the scene's boundaries? In Figure 3-3, I noticed that the output of Y(=7.08) seems not very accurate compared with the other aligned ones.
2.  The idea is very interesting. However, I still need to point out that the runtime of the model inference is not efficient enough for the practical problems (planning, localization, etc.) discussed in the paper.
3.  The phenomenon that the MLLM has the capability to perceive the object's 3D position is interesting. However, the pipeline requires external models like SAM to first get the masks/other auxiliary markers to highlight the target object. Step back and think about the whole pipeline, it only proves that the MLLM can do a very simple task--- given a cartesian coordinate system represented by 3 orthogonal axes and a 3D point, MLLM is asked to output the coordinates of this 3D point.  I think it would be interesting to experiment with the simplest case mentioned above to see if the MLLM can output the arbitrary 3D point's coordinates and how accurate it is.  So that we can better understand if the semantics in the 3D scene help in the localization process.
4. It would be more interesting if more types of coordinate systems could be tested. For example, polar coordinate systems and cylindrical coordinate systems. We can also test on some self-defined coordinate systems, for example, with log-scaled axes.
I am curious about how MLLM possesses such spatial sensing capability without direct training. I wonder if there are many images of cartesian frames contained in the training data.

**Questions:**

Please refer to the weakness part.

---

> ### Author Response · Authors · 2024-11-20
>
> We appreciate the reviewer’s thoughtful feedback
>
> ### w1: How are the 3D axes set? Does the origin always lie in the same corner of the scene? What if the axes are not well aligned with the scene's boundaries? In Figure 3-3, I noticed that the output of Y(=7.08) seems not very accurate compared with the other aligned ones.
>
> The 3D axes are aligned with a principal component of the input point cloud. On the scannet, if the axes are not well aligned with the scene's boundary, the performance will significantly drop.
>
>
> ### w2: The idea is very interesting. However, I still need to point out that the runtime of the model inference is not efficient enough for the practical problems (planning, localization, etc.) discussed in the paper.
>
> The runtime is indeed a limitation for practical problem-solving. We will add a discussion of the limitations in the final paper.
>
>
>
> ### w3: Ball positioning experiments
>
> We conduct the point positioning task as shown in [Figure 1](https://github.com/Anonymous123-ctrl/3DAxisPrompt540/blob/main/point_position.png). The ball position can be acquired by direct simple question "Can tell me the position of the ball from the images?" and the input prompted images. We test about 20 times and the results are as follows.
> | 3D Axis |Mark Type | Prompt Elements  | Metric|
> |---------|-----------|-------------------|---------|
> Scannet    |   - | -          |  To center
> ~   | 3D Mark | Mark                                                |   N/A  |
> Yes| baseline  |  -                                                       | 0.391 |
> Yes| 2D Mark | Mark+2D contour (colors) + CoT | 0.219 |
> Point Positioning    |   - | -          |  To center
> Yes| 2D Mark |  Mark+2D contour (colors) + CoT |  0.184
>
> From the results, we can see that the ball localization is more accurate than the object localization.
>
> Furthermore, we from all the other experiments and the investigation. We find that the MLLM can utilize these localization abilities. In other words, the localization abilities help promote the 3D grounding abilities, as shown in the comparison experiments with LLM-Grounder.
>
> ### w4: other scaled axis
> We are working on these experiments and results will be provided soon.

---

### Official Review · Reviewer_yDrG · 2024-11-01

**Soundness:** 2
**Presentation:** 1
**Contribution:** 1
**Rating:** 3
**Confidence:** 4

**Summary:**

The paper introduces 3DAxisPrompt, a new visual prompt method to enhance 3D understanding and reasoning in MLLMs. It explores various visual prompts, including 3D axes, tri-view images, 2D contours, and 3D edge points. Experiments are conducted on several self-designed tasks, such as indoor/outdoor localization, route planning, robot action prediction, and coarse object generation.

**Strengths:**

The paper explores a range of visual prompts (coordinate axis, masks, bounding boxes, marks, color highlights) for 3D grounding and reasoning, tested on several self-designed tasks like indoor localization, robot action prediction, and route planning.

**Weaknesses:**

1. The term "3D grounding ability" seems overstated, as the paper only assesses its own indoor localization tasks without referencing established benchmarks like ScanRefer or Referit3D.

2. The paper lacks comparisons with existing methods on standard benchmarks, which are essential to reveal its limitations or gaps relative to current learning-based and prompt-based 3D grounding approaches, such as LLM-Grounder [1]. By only testing its performance in isolation, the study fails to demonstrate an advantage over other methods. It would be more convincing if the authors:
- Provide results comparison on comprehensive, widely accepted benchmarks (e.g., compare directly with public 3D grounding, 3D detection, or 3D QA datasets),
- Demonstrate real-life applications, as seen in works like PIVOT, by applying the method to practical scenarios.

3. The paper lacks an in-depth analysis of view selection in real-world settings. For instance, real-world 3D axes often cannot be perfectly aligned with point clouds. Is there a systematic study on this effect or a solution for optimal view selection? Additionally, could zooming in and applying axis prompts improve localization accuracy? Addressing these technical details would enhance the paper’s guidance on effectively using axis prompts.

[1] LLM-Grounder: Open-Vocabulary 3D Visual Grounding with Large Language Model as an Agent. ICRA 2024

**Questions:**

See the weakness.

---

> ### Author Response · Authors · 2024-11-19
>
> We thank the reviewer for the insightful feedback.
>
> ### w1: comparison on widely accepted benchmarks
>
> We add more comparision experiments with LLM-Grounder on the ScanRefer dataset as follows:
>
> |Method | Visual Grounder | LLM Agent | Acc@0.25 | Acc@0.5|
> |--------|-----------------|------------|------------|---------|
> ScanRefer | - | - | 35.2 | 20.7
> LERF | LERF | - | 4.5 | 0.3
> LLM-Grounder | LERF | GPT-4 | 7.0 (+2.5) | 1.6 (+1.3)
> Ours | LERF + 3DAxisPrompt | GPT-4o | 9.1 (+4.6) | 3.1 (+2.8)
>
> Specifically, we replace the LLM Grounder with the proposed 3DAxisPrompt+GPT-4o. Each QA, the Input Text prompt of LLM-Grounder and the observation images are set as input to GPT-4o, the rest remains the same.
>
> From the results, we can see that the 3D AxisPrompt significantly improves the performance of the grounding performance of LERF and LLM-Grounder.  This shows that the 3DAxisPrompt plus the text prompt of LLM-Grounder can significantly promote grounding abilities.
>
> The reason that we choose 3D localization is that the localization ability is only provoked by the proposed 3DAxisPrompt and we think it can represent the 3D grounding abilities in some way. The LLG-Grounder is very  inspiring to us. Therefore, we add more comparision results follows the LLM-Grounder
>
> ### w2: in-depth analysis
> It is important to align the x or y axis with the main direction of the scene. On scannet, the main direction is the x or y of the floor. In fact, we develop a simple algorithm in our observation image rendering tool to determine the direction by calculation the Principal component of the input point cloud, code will be publised if accept.
>
> There is a systematic study on the view selection. However, to eliminate the view selection, we choose to render multiview images and set the series of images as the input visual prompt, as shown in [Folder](https://github.com/Anonymous123-ctrl/3DAxisPrompt540/tree/main/pipeline/ObservationImagesby3DAxisPrompt).
>
> Zooming in will improve localization accuracy. Also, some text prompt regarding the coarse position description in the instruction will also improve the performance.
>
> We will add a comprehensive conclusion of details like these in the final paper.

---

> > ### Comment · Reviewer_yDrG · 2024-11-25
> > **Thanks for the author rebuttal.**
> >
> > Thank you for the author's rebuttal. However, they did not resolve my concerns:
> > 1. The paper lacks comparisons on several other benchmarks, and the claims of 3D grounding do not seem appropriate.
> > 2. The study does not ablate the effects of view selection.
> > Given these reasons, I believe the paper remains below the acceptance threshold.

---

### Official Review · Reviewer_91eh · 2024-11-03

**Soundness:** 2
**Presentation:** 3
**Contribution:** 1
**Rating:** 3
**Confidence:** 2

**Summary:**

The authors proposed 3DAxisPrompt, which is a prompting method for 3D grounding questions for multimodal large language models (MLLM). The prompting basically consists of two parts: 1) add the 3D axis to the point cloud and render observation images from multiple views of the point data; 2) add masks/boundboxes generated from SAM in the rendered image to provide explicit geometric priors.The authors test the proposed prompting on GPT-4o (ChatGPT interface) on 20 scenes from each test dataset (ShapeNet, ScanNet, FMB and nuScenes) and show that such prompting can help GPT-4o interpret 3D space.

**Strengths:**

- The paper is well-written and lucidly presented.
- The related work section provides a comprehensive overview of MLLMs and prompting techniques.
- The prompting method is straightforward and easy to apply.
- The authors demonstrate the efficacy of the 3DAxisPrompt on a limited set of scenarios through indoor localization/route planning on ScanNet, outdoor localization on nuScenes, and robot action prediction on FMB dataset.

**Weaknesses:**

- Limited contribution: The primary contribution of this paper is the 3DAxisPrompt prompting method for MLLM, along with testing GPT-4o on such prompts. The prompting method is straightforward and simple. While it is generally beneficial to present a simple method, I anticipate an in-depth analysis demonstrating its essentiality and key to resolving the problem. However, the paper lacks scientific investigation into why MLLM can perform 3D grounding with the proposed prompting and does not provide a novel perspective on the problem. The paper’s structure resembles a technical report rather than a conference contribution.
- Evaluation issues:
    - Quantitative evaluation of adding 3D axis: The proposed 3DAxisPrompt comprises two parts, but there is no quantitative evaluation of the first part (adding 3D axis) presented in the paper.
    - Baseline methods: There is no baseline shown in table 1 and 2 and it is hard to interpret the numbers in the tables.
    - Scientific validity and significance: The evaluation of the proposed 3DAxisPrompt is not scientifically valid or significant. The paper evaluates the proposed 3DAxisPrompt on only a single pre-trained MLLM model (GPT-4o) on a very limited set of scenes. It is challenging to ascertain the scientific significance of the results obtained from such a limited set of scenes. Furthermore, without testing on other MLLM models, it remains uncertain whether the proposed prompting can generalize.

**Questions:**

Typo:
- L349: rend -> trend.

---

> ### Author Response · Authors · 2024-11-19
>
> Thanks for the constructive suggestions.
>
> ###  w1:Limited contribution: However, the paper lacks scientific investigation into why MLLM can perform 3D grounding with the proposed prompting and does not provide a novel perspective on the problem. The paper’s structure resembles a technical report rather than a conference contribution.
>
> We have to restate that we think the idea is worth cherishing. The 3D grounding ability is indeed promoted by a simple and easy-to-deploy prompt. Also, we add more comparision experiments with LLM-Grounder on the ScanRefer dataset as follows:
>
> |Method| Visual Grounder | LLM Agent | Acc@0.25 | Acc@0.5|
> |---------|--------------------|--------------|-------------|-----------|
> ScanRefer | ~ | ~ | 35.2 | 20.7
> LERF | LERF | ~ | 4.5 | 0.3
> LLM-Grounder | LERF | GPT-4 | 7.0 (+2.5) | 1.6 (+1.3)
> Ours | LERF + 3DAxisPrompt | GPT-4o | 9.1 (+4.6) | 3.1 (+2.8)
>
> Specifically, we replace the LLM Grounder with the proposed 3DAxisPrompt+GPT-4o. Each QA, the Input Text prompt of LLM-Grounder and the observation images are set as input to GPT-4o, the rest remains the same.
>
> From the results, we can see that the 3D AxisPrompt significantly improves the performance of the grounding performance of LERF and LLM-Grounder.  This shows that the 3DAxisPrompt plus the text prompt of LLM-Grounder can significantly promote grounding abilities.
>
> Also, we consider the visual prompting process as introducing lost 3D information when capturing the 2D observation images. Following this thought, we conduct comprehensive investigation on the potential prompt format as in Section 3.2 and 3.3 and the supplement section. As for why the MLLM can perform 3D grounding with the prompting. First, we conduct the ablation study and find out the axis, ticks, and labels are all important in promoting 3D grounding. Based on the ablation study and the investigation on the potential prompt format, we try to conclude our perspective on the 'why' question, as follows:
> 1. Initial grounding abilities are found in the LMM, such as knowing the cup is on the desk, from the 2D images. However, this grounding ability is limited to the relative position relationship.
> 2. By adding the proposed visual prompts, we think it acts like a ruler or anchor for the scene. Thus, based on the relative position relationship sensing abilities, MLLM can find the object relative position to the axis, thus producing the 3D localization results.
> 3. To support our findings, it can be seen from the ablation study that the 3DAxisPrompt totally fails when removing the ticks of the axis. This shows that the numbers that indicate the absolute position relationship is essential and MLLM relies on the anchor to sense the 3D position from the 2D observation images. Also, when other elements of the axis are removed, 3DAxisPrompt can still work but the performance drops a little bit (3.5%). This also implies the importance of ticks.
> 4. In conclusion, the MLLM provoke the 3D grounding (esspecially 3D localization) based on the absolute position achored by the axis and the relative position relationship sensing.
>
> ### w2:  Evaluation issues:
>
> We enrish the Table 1 in the paper as shown below. We add the first part (adding 3D axis). Also, to better interpret the numbers in the table, we add the results of only using 3D axis without additional marker as baseline. As for no axis, the MLLM fails to provoke 3D localization.
> |3D Axis |Mark Type | Prompt Elements  | Scannet | Scannet|
> |---------|-------------|----------------------|-----------|----------|
> ~  |   - | -          |  To center | To bbx
> No  | 3D Mark | Mark                                                |   N/A  | N/A
> Yes| baseline  |  -                                                       | 0.391 | 0.296
> Yes| 3D Mark | Mark                                                | 0.333 | 0.216
> Yes| 3D Mark  | Mark+OBB                                     | 0.350 |0.231
> Yes| 3D Mark  | Mark+AABB (red)                          |0.376 |0.219
> Yes| 3D Mark  | Mark+AABB (colors)                     |0.311  |0.207
> Yes| 3D Mark  | Mark+3D edge points                   |0.305 |0.205
> Yes| 2D Mark |2D contour (colors)                         | 0.320|0.175
> Yes| 2D Mark | Mark+2D contour (colors)             | 0.271 | 0.138
> Yes| 2D Mark | Mark+2D contour (colors) + CoT | 0.219 | 0.115
>
> ### w3: Scientific validity and significance
>
> we add more comparison experiments with LLM-Grounder on the ScanRefer dataset as follows:
>
> |Method | Visual Grounder | LLM Agent | Acc@0.25 | Acc@0.5|
> |--------|-----------------|------------|------------|---------|
> ScanRefer | - | - | 35.2 | 20.7
> LERF | LERF | - | 4.5 | 0.3
> LLM-Grounder | LERF | GPT-4 | 7.0 (+2.5) | 1.6 (+1.3)
> Ours | LERF + 3DAxisPrompt | GPT-4o | 9.1 (+4.6) | 3.1 (+2.8)
>
> ### w3: Typo:
>
> Truly sorry for the mistakes, typos will be rechecked.

---

> > ### Comment · Reviewer_91eh · 2024-11-25
> >
> > Thanks the authors for the rebuttal, but I think my concerns on the evaluation issues (especially no baseline in table 1 and 2, scientific validity and significance) and the limited contribution were not resolved. Thus I keep my original rating.

---

### Official Review · Reviewer_i659 · 2024-11-04

**Soundness:** 2
**Presentation:** 2
**Contribution:** 2
**Rating:** 3
**Confidence:** 5

**Summary:**

This paper focuses on significantly enhancing the understanding of 3D spatial information by mature LLMs like GPT-4-O through the use of different forms of prompts.

**Strengths:**

This paper focuses on significantly enhancing the understanding of 3D spatial information by mature LLMs like GPT-4-O through the use of different forms of prompts.

**Weaknesses:**

This paper focuses on significantly enhancing the understanding of 3D spatial information by mature LLMs like GPT-4-O through the use of different forms of prompts. However, I have the following questions: 1. To what extent does this method improve some 2D LLMs, such as LLAVA-NEXT? 2. In addition to the benchmarks listed in the paper, can this method be applied to more benchmarks? I need more confidence and evidence of improvements on widely recognized benchmarks. 3. This method seems a bit too simplistic; please restate its innovativeness and necessity, as well as how it differs from similar approaches. If necessary, more visualizations can be provided.

**Questions:**

This paper focuses on significantly enhancing the understanding of 3D spatial information by mature LLMs like GPT-4-O through the use of different forms of prompts. However, I have the following questions: 1. To what extent does this method improve some 2D LLMs, such as LLAVA-NEXT? 2. In addition to the benchmarks listed in the paper, can this method be applied to more benchmarks? I need more confidence and evidence of improvements on widely recognized benchmarks. 3. This method seems a bit too simplistic; please restate its innovativeness and necessity, as well as how it differs from similar approaches. If necessary, more visualizations can be provided.

---

> ### Author Response · Authors · 2024-11-19
>
> We are truly sorry for the delayed reply.
>
> ### w1: To what extent does this method improve some 2D LLMs, such as LLAVA-NEXT? In addition to the benchmarks listed in the paper, can this method be applied to more benchmarks? I need more confidence and evidence of improvements on widely recognized benchmarks.
>
> We are trying our best to conduct comparision experiemtns with LLM-Grounder on ScanRefer dataset, the comparision experiments also mentioned by other reviewers. The workload is massive.
>
> Now we provide the following additional results on ScanRefer:
>
> Method | Visual Grounder | LLM Agent | Acc@0.25 | Acc@0.5
> ----------|-------------------|--------------|-------------|-----------
> ScanRefer  | - | -  |  35.2 | 20.7
> LERF | LERF | - | 4.5 | 0.3
> LLM-Grounder | LERF | GPT-4 | 7.0 (+2.5) | 1.6 (+1.3)
> Ours | LERF + 3DAxisPrompt | GPT-4o | 9.1 (+4.6) | 3.1 (+2.8)
>
> Specifically, we replace the LLM Grounder with the proposed 3DAxisPrompt+GPT-4o. Each QA, the Input Text prompt of LLM-Grounder, and the observation images are set as input to GPT-4o; the rest remains the same, as shown in [Figure 1](https://github.com/Anonymous123-ctrl/3DAxisPrompt540/blob/main/vs_llm-Grounder.pdf). We add an anonymous link for display.
>
> From the results, we can see that the 3D AxisPrompt significantly improves the grounding performance of LERF and LLM-Grounder.  This shows that the 3DAxisPrompt plus the text prompt of LLM-Grounder can significantly promote grounding abilities.
>
> We are still working on more comparison experiments.
>
> ### w2: This method seems a bit too simplistic; please restate its innovativeness and necessity, as well as how it differs from similar approaches. If necessary, more visualizations can be provided.
>
> Our initial attempt is to expand the 2D visual grounding abilities to 3D visual grounding abilities in the vision-language model (VLM) by the proposed visual prompts. We choose the 3D localization ability of the 3D grounding because it is challenging for VLM (such as GPT-4o) to invoke 3D grounding ability (especially 3D localization)  without finetuning or network changes. We think the 3D localization from 2D images is a strong point to support our statement "promote the 3D grounding ability".
>
> Furthermore, the visual prompts can be seen as introducing more 3D information lost when capturing the observation images from the 3D scene. Following this thought, we designed the visual prompt scheme (3DAxisPrompt).
>
> The innovation is that the vision language model (such as GPT-4o) struggles to process the 3D grounding abilities from 2D images without finetuning or modifying the network. Notably, we are not able to play with the mega model like the GPT series. However, by preprocessing the input images with the visual prompt scheme, the model can show more powerful 3D grounding abilities (especially the 3D localization through 2D images). We think this significantly makes these mega models like GPT-4o more powerful.
>
> The necessity is that we consider the Large Multi-modal Model (LMM) as an interaction tool that can interrelate humans and their surrounding environments. GPT-4o, as one of the most powerful AI (LMM), should be able to possess the potential to become the interaction tool. However, 3D localization from 2D images or other problem-solving abilities necessitate this localization ability, which is challenging for GPT-4o right now. We think it is necessary to empower the LMM with more powerful 3D grounding abilities so that it can help solve more tasks based on the promoted 3D grounding abilities.
>
> We provide more visualized pipeline as shown in [Figure 2](https://github.com/Anonymous123-ctrl/3DAxisPrompt540/blob/main/pipeline.pdf) and more intermedia results in the [Folder](https://github.com/Anonymous123-ctrl/3DAxisPrompt540/tree/main/pipeline)
>
> [1] LLM-Grounder: Open-Vocabulary 3D Visual Grounding with Large Language Model as an Agent. ICRA 2024

---

> > ### Comment · Reviewer_i659 · 2024-11-26
> >
> > Thank you for the author's rebuttal. However, they did not resolve my concerns. Thus I keep my rating.

---

### Meta-Review · Area_Chair_hVQH · 2024-12-19

**Metareview:**

**Summary**

The paper proposes a method (3DAxisPrompt) to prompt multimodal large language models (MLLMs) and investigate the ability of MLLMs to tackle 3D grounding and reasoning tasks.  3DAxisPrompt works by 1) providing 3D coordinate axis together with rendered scene to MLLMs, 2) using masks with Segment Anything with Set-of-marks prompting.  The proposed method is evaluated on various tasks such as localization, route planning, and action prediction.  Ablations are also conducted on how to encode axis information, number of  multi-view images, etc.

**Strengths**

Reviewers appreciated the following:
1. The investigation of prompting MLLMs for 3D tasks is an interesting direction that can be applied to different tasks [HR2p]
2. Experiments are conducted on a variety of tasks and datasets [91eh, yDrG, HR2p]
3. Proposed method is easy to understand and apply [91eh, HR2p]

**Weaknesses**

The main concern from reviewers are weaknesses in the evaluation.  Reviewers noted the following:
1. Limited evaluation
   - Only GPT-4o is evaluated, additional MLLMs (e.g. LLAVA-NEXT) should be considered [i659, 91eh]
   - Lack of baselines and comparisons against existing methods [91eh,yDrG]
   - Limited ablations [91eh]
2. Lack of evaluation on standard benchmarks [i659,yDrG]
3. Lack of in-depth analysis [91eh,yDrG]

**Recommendation**

Due to reviewer concerns, the AC finds the work not ready for publication at ICLR and recommends reject.

**Additional Comments On Reviewer Discussion:**

The submission received three rejects [yDrG, 91eh, i659], and one borderline accept [HR2p].

In response to reviewer concerns, authors provided additional comparisons during the author response period.  However, the manuscript itself was not revised.  In addition, the comparisons did not necessarily respond to questions that were asked by reviewers (e.g. R-i659 had a question regarding performance with LLAVA-Next, experimental results for LERF was provided).

Due to the above, reviewers indicated that their concerns remain unaddressed.  One reviewer [yDrG] lowered their rating from 5 to 3 after the author response period.

---

### Decision · Program_Chairs · 2025-01-22

Reject